# Combination of Cross-Correlation-Based Analysis and Ultrasonic GW Tomography for Barely Visible Impact Damage Detection Preliminary Assessment

Monica Ciminello [1,*] , Natalino Daniele Boffa [2], Salvatore Ameduri [1] and Ernesto Monaco [3]

1   Department of Adaptive Structures, C.I.R.A. Italian Aerospace Research Center, Via Maiorise snc,
    81043 Capua, Italy; s.ameduri@cira.it
2   ANAS S.p.A., Italian National Autonomous Roads Corporation—General Directorate, Via Monzambano, 10,
    00185 Rome, Italy; n.boffa@stradeanas.it
3   Department of Industrial Engineering, University degli Studi di Napoli Federico II, Via Claudio 21,
    80125 Naples, Italy; ermonaco@unina.it
*   Correspondence: m.ciminello@cira.it; Tel.: +39-0823623535

**Abstract:** Statistical based reconstruction methods and signal processing tooling techniques are implemented and used to detect delaminations or debondings within composite complex items with very high precision. From the literature, it appears that although a single procedure for the estimation of structural health is a fast solution, multiple analyses based on different reconstruction methods or different damage parameters are the way to achieve maturation assessments of the methodology. This highlights the fact that the hardware and software parts of an SHM system need two different assessment and maturation ways. This work focuses on the software part by proposing a way to start assessing the outcomes. In this paper, the damage detection and localization strategy in CFRP plate-like structures with elastic guided waves excited and acquired with a circular array PWAS network is considered. Previous outcomes are compared by new analyses using a new post-processing approach based on a cross-correlation-based technique in terms of the BVID (Barely Visible Impact Damage) surface position and its center of mass. The advantage of this specific study is hopefully to enable confidence in the transition from R&D to field implementation. In addition, this work tries to evidence an improvement in terms of cost efficiency and reduced complexity while maintaining the same accuracy.

**Keywords:** composite structures; structural health monitoring; cross-correlation analysis; guided waves; piezoelectric sensors

## 1. Introduction

Structural Health Monitoring (SHM) technology offers a new approach to interrogate the integrity of structures in real-time or on demand without physically disassembling the structures. The objective is to enhance operating safety, increase availability and reduce direct operating costs. A huge amount of scientific studies show how the correct implementation of SHM can have a positive impact on the life-cycle cost of a structure, and therefore presents a positive cost/benefit ratio [1].

Unfortunately, the implementation of SHM systems and their commercial deployment is delayed by the lack of maturation. Indeed, no formal methodology has yet been developed to address SHM system maturation despite the fact that it is a critical step to guarantee its reliability and therefore its certification, as pointed out in recent publications [2–4]. Hence, as for NDE (non-destructive evaluation) systems, a specific Verification and Validation (V&V) procedure for the maturity assessment of SHM systems needs to be realized [5].

The designing and implementing of a cost-effective SHM system constitute a process that must be carried out following a logical sequence of analysis steps and decisions that respond to the needs of all industry parties. Here is a short recap:

Identification of a region of interest needing monitoring. This step might seem trivial but is indeed a very important first step. Before considering a structural health monitoring system, it is important to consider if a specific structure will really benefit from it.

Opportunity analysis is the second phase. The SHM system designer and the owner must jointly identify the risks, uncertainties and opportunities associated with the specific structure. The risk analysis will lead to a list of possible events and degradations that can possibly affect the structure.

Responses selection comes soon after. For each of the retained risks, it is fundamental to associate one or several responses that can be observed directly or indirectly. For example, corrosion produces a chemical change, but also a section loss. This is very important to select sensors with the appropriate specifications.

Then there is the design of the SHM system and the selection of appropriate sensors. The goal is to select the sensors that have the appropriate specifications to sense the expected responses and are appropriate for installation in the specific environmental conditions and under the technical constraints found in the structure. It often makes sense to include sensors based on different technologies to decrease the system's redundancy. On the other hand, having too many data acquisition systems will increase the system cost and complexity, so a good balance is required.

Calibration and Data Acquisition and Management

These are the operational parts of the process. The result of this step is a database of measurements and a log of events.

The final step is the Data Assessment. The output of this step is a series of alerts, warnings and periodic reports. But data assessment can also be achieved by improving data management by reducing the complexity and computational time effort.

Focusing on guided wave based approaches, being of specific interest to this work, there are many papers published concerning the successful detection and localization of damage by using various DIs and mathematical functions based on time reversal, linear sampling, artificial intelligence, etc. [6–13].

Among the most popular reconstruction methods, which rely on consolidated tomographic algorithms, the Reconstruction Algorithm for Probabilistic Inspection (RAPID) is recognized to be one of the most verified and validated [14]. It is worth mentioning that several authors have proposed the implementation of a modified version to make it suitable for the majority of different scenarios. That is the proper way to estimate the assessment of a methodology.

This algorithm commonly requires that a healthy baseline is acquired in the absence of any damage in the early life of the structure, which is stored and later compared to the subsequently acquired data during the operational life of the system. Then, a tomography of the structure can be properly built, further processing the algorithm output, which consists of a matrix whose components are the damage probabilities for each of the pixels of interest in the structure [14].

Several authors have then proposed the RAPID algorithm without the need for a baseline comparison [15–17]. However, the implementation complexity sharply grows, requiring, in most cases, an a priori and complete knowledge of the guided waves (GW) propagation physics. Notwithstanding the easy implementation, it also presents some weaknesses that, in the authors' opinion, have not been deeply considered in the literature.

Liu et al. [18] partially addressed the so-called uneven sensing network density by subtracting a "compensating image", which identifies the probability distribution tomography when the sensing paths have the same weights. Although there was no appreciable performance improvement in terms of damage localization between the compensating and standard method, the authors managed to satisfactorily increase the tomography resolution, reducing the extent of the estimated high probability area.

Azuara et al. [19] also proposed a variation for solving the uneven density of the sensing network. They employed a shape function to diminish the probability around the sensing path intersection points. The final tomography was greatly improved in terms of both contrast and accuracy. Yet, the proposed technique also introduced an extra subjective parameter to define the area of influence of the shape function.

Consequently, even if the aforementioned solutions provide satisfactory results, they also add some input parameters which may decrease the objectivity of the algorithm.

Most of the existing vibration-based SHM methods could be classified into two different approaches: global approaches and local approaches. In the global approaches, the goal is to monitor the health of the entire structure. However, for many large systems, global monitoring is not practical due to the lack of sensitivity of global features regarding local damages, inaccuracies of developed models or the high cost of sensing, cabling and computational operations. On the other hand, local SHM methods are focused on evaluating the state of reduced parts within the entire structures, based on substructuring methods. This approach aims to overcome the global method problems, dividing the whole structure into substructures and analyzing each one individually.

*Scope of This Work*

The results provided in this paper are obtained for a rather simple specimen, but it might be noted that the scope of this work relies on the development of a relatively simple yet robust approach for damage localization and visualization.

The work starts from the idea that hardware and software parts of an SHM system need two different assessment and maturation ways and in this case, the activity focuses on the software part by proposing a way to start assessing the outcomes.

This work deals with the combination of a cross-correlation-based analysis and ultrasonic guided wave tomography for BVID detection. In this case, the ultrasonic guided wave tomography approach, previously published by the authors [20–23], is tested and evaluated through the use of the new signal processing tool based on cross-correlation analysis [24–26] already tested for static strain data and herein applied for the first time on guided wave signals. The modified version is applied on a single test case as preliminary results to compare variations in cost-efficiency improvement and complexity reduction as a way for methodology assessment and maturation.

For the sake of clarity, it is assumed that the circular sensor network layout and the originally featured signal extraction steps are optimized, so they do not need any improvements.

Indeed, it is demonstrated that two opportunities are available to improve the localization performance when using a sensor network and guided waves: the first one is a localization strategy by using the ellipses method that seems to give good results [6]. The second one is to restrict the area of the impact research in reasoning by cluster, which provides a better resolution [6]. The circular PZT layout herein proposed belongs to this second approach.

What comes after is a short description of the monitored structure in Section 2, and a sensor layout presentation in Section 3. Section 4 presents the proposed assessment procedure which is then applied to a single BVID case study. From Sections 4 and 5, the assessment process is applied through this case study and allows for the evaluation of the performance of SHM algorithms.

## 2. Test Article Description

The objective of SHM technology is to demonstrate the possibility to carry out robust monitoring devices, based on ultrasonic sensors, able to detect in service damages. For these purposes, a full scale ground test with 4.5 m span outer wing demonstrators has been tested to validate the structural monitoring system functionality. SHM implementation and testing have been performed by Federico II University of Naples (UniNa) as leader of the SHM scenario within the EU-funded SARISTU project, in collaboration with all the partners [1,20,21]. The tested wing box demonstrator consisted of a main box realized

as a composite of pristine parts in terms of panels, stringers, ribs, and rear & front spars (Figure 1).

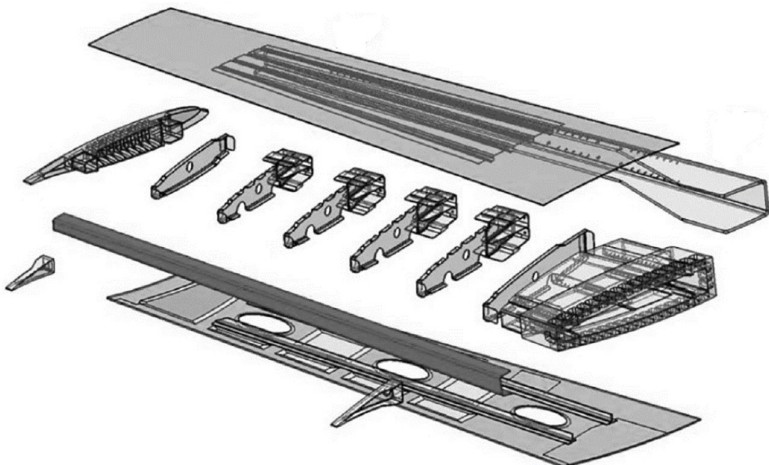

**Figure 1.** Full scale wing box ground demonstrator main components exploded view.

Before wing box demonstrator assembly, the wing box lower panel (LWP) material has been modelled and sensorized for SHM system implementation. The sensor distribution was defined by taking into consideration the potential impact locations settled in the project phase. The preliminary sketch of a possible sensor network layout is provided in Figure 2. The regions of interest for such a large wing panel have been indicated by a red rectangle, each labelled by a capital letter. Coherently with the main principle of the tomography, each region is supposed to be monitored by a specific 1-D or 2-D sensor layout geometry according to the target (skin-spar bonding line or surface impact). In this work, an optimized circular 2-D geometry is considered for BVID.

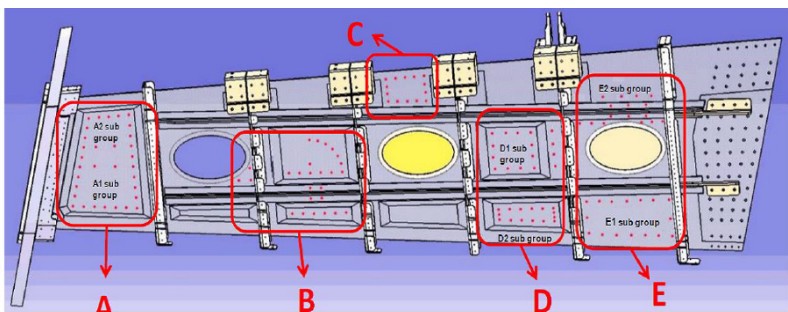

**Figure 2.** LWP sensor tags indicated by red dots.

The material inhomogeneity, anisotropy and the multi-layered construction of composite materials lead to significant dependence of wave modes on laminate layup configurations, direction of propagation, frequency and interface conditions.

A preliminary tapered test panel has been built in order to test the three different lay-ups of composite plates characterizing the wing panel; the panel was divided into three bays (6, 8 and 10 mm thick) and the activities presented within this work have been carried out on one of the three (the 6 mm) in order to validate the numerical and experimental methodologies and technologies before upscaling them to the full wing panel.

The panel is composed of 12 laminae of three different kinds of prepreg oriented along several directions. The characteristics of the plies, as provided by the manufacturer, are summarized in Table 1.

**Table 1.** Initial characteristics of the materials.

| Material | Density | E1 | E2 | G12 = G23 = G31 |
|---|---|---|---|---|
| 5harness | 1.77 g cm$^3$ | 65,000 MPa | 65,000 MPa | 3600 MPa |
| Biaxial | 1.79 g cm$^3$ | 81,000 MPa | 81,000 MPa | 4100 MPa |
| Uniaxial | 1.79 g cm$^3$ | 152,000 MPa | 8800 MPa | 4100 MPa |

## 3. Sensor Network Layout and Signal Acquisition

In the experimental tests, a calibrated pre-loaded spring gun impact machine has been used with a 1-inch striker (Figure 3a). Then, the panel is subjected to low velocity impact (85 J) damage tests, with the damaged area corresponding approximately to a 26 × 25 mm ellipse.

The PWAS (Piezoelectric Wafer Active Sensors) tested were DuraAct P-876.SP1 with dimension "16 × 13 × 0.6 mm", active area of 0.64 cm$^2$ and a mass of 0.5 g. DuraAct patches (commercial rectangular and customized circular shape) employed (see Figure 3b) were both based on a thin piezoceramic foil between two conductive films, all embedded in a ductile composite polymer structure. In this way, the brittle piezoceramic is mechanically pre-stressed and electrically insulated, which makes the transducers more robust and therefore applicable on curved surfaces. Thirteen transducers have been permanently bonded on the structure by employing a vacuum-based secondary bonding procedure of common use by the aircraft industry. The radial sensors pattern is adopted for optimal monitoring of the enclosed surface of the plate. A fourteenth disk is installed at the circle's center to perform a propagation velocity analysis. The overall configuration of the instrumented panel is shown in Figure 3b.

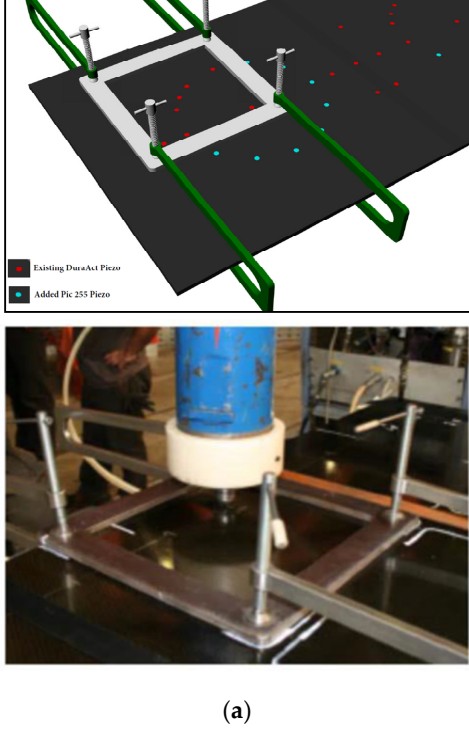

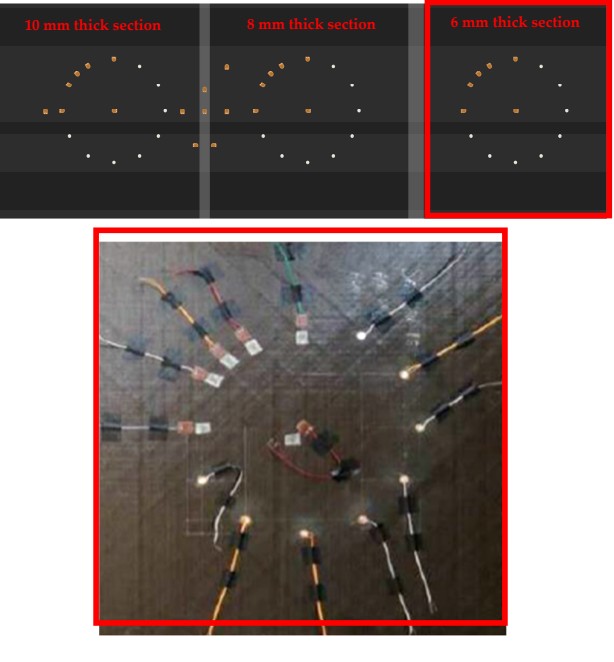

(**a**)                                                                                                                              (**b**)

**Figure 3.** *Cont.*

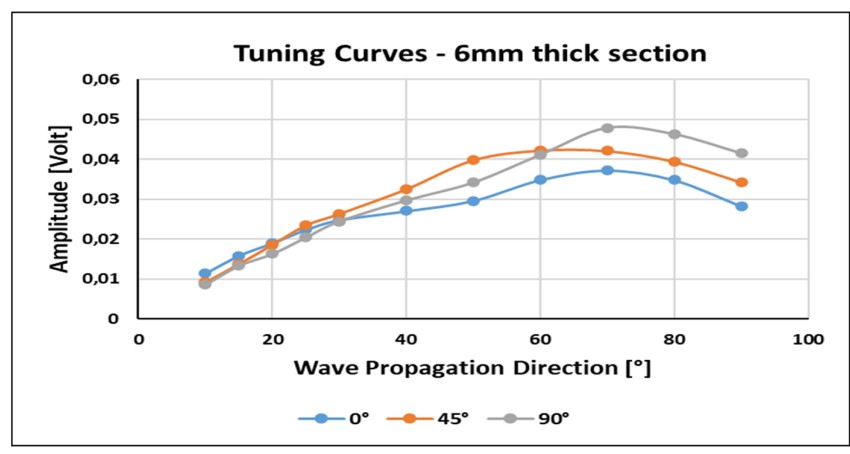

(**c**)

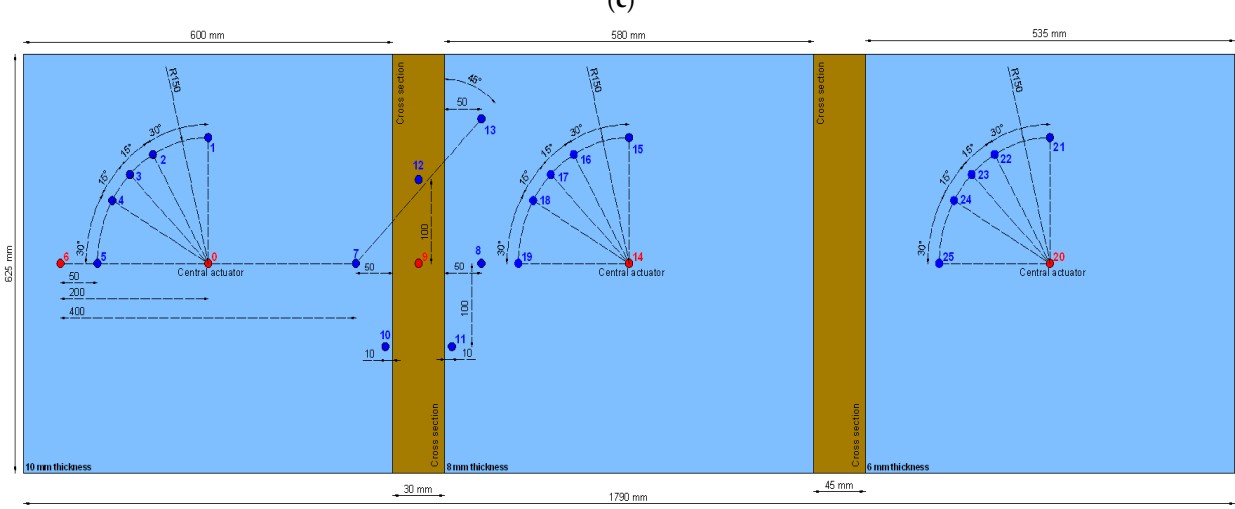

(**d**)

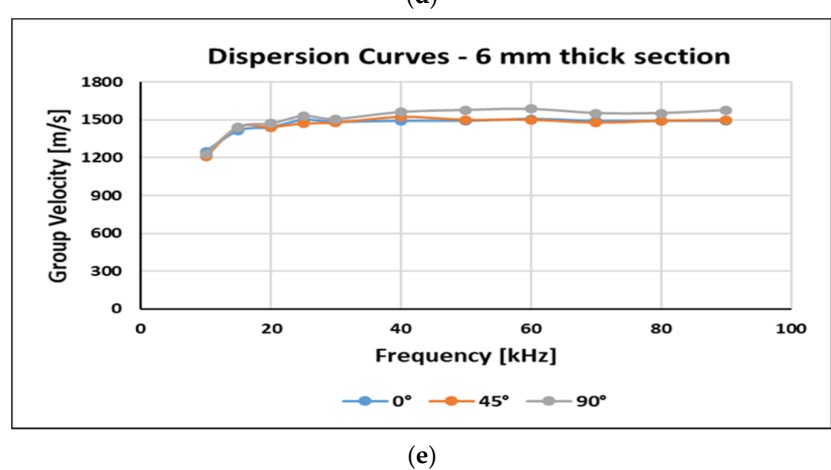

(**e**)

**Figure 3.** (**a**) Impact test setup; (**b**) sensors configuration; (**c**) experimental tuning curves for the "6 mm" thick bay; (**d**) tapered test panel sketch; (**e**) dispersion curves for the "6 mm" thick bay.

The elastic waves are launched by broadband transducers located on the surface of the structure. In this regard, the PWAS sensor satisfies most of these characteristics and the most widely used sensor for actuator or sensor applications having dimensions $16 \times 13 \times 0.6$ mm, active area of $0.64$ cm$^2$ and a mass of $0.5$ g has been tested. The source signal, generated by (HP/Agilent 33120A), consists of a 4.5 sine cycles signal with 10 V peak-to-peak tension Hanning window and a 60 kHz central frequency, adopted after an

experimental PWAS guided wave tuning along 0, 45 and 90 degree propagation directions (Figure 3d). The signal amplitude and the influence of the edge reflections have been considered as key parameters in the central frequency choice. An amplifier has been used to burst the PZT sensors with up to 80 V peak-to-peak signal in the experiments. The ultrasonic signal has been digitized and recorded directly in a four-channel digital oscilloscope with 100 MHz sampling rate (Agilent InfiniiVision DSO7104A). The digital ultrasonic signals are then downloaded to a personal computer and post processed. In the preliminary analysis, dispersion curves of 6 mm thick bay of tapered panel have been calculated along 0, 45 and 90 degree propagation directions (Figure 3e).

Changes in the measured dynamic response of the structure were analyzed to reveal the presence of damage. Impact delamination causes Lamb waves to propagate in a different thickness condition; thus, this has an effect on wave propagation velocity. A shift in the arrival time can be observed on the raw signal (Figure 4a). To check the influence of a defect, the main effect seen will be an attenuation or an absorption of wave energy (Figure 4b). To quantify this effect and maybe correlate it with the size of the delamination, the amplitude must be precisely determined. The changes in the amplitude of wave packages can be used to detect damage.

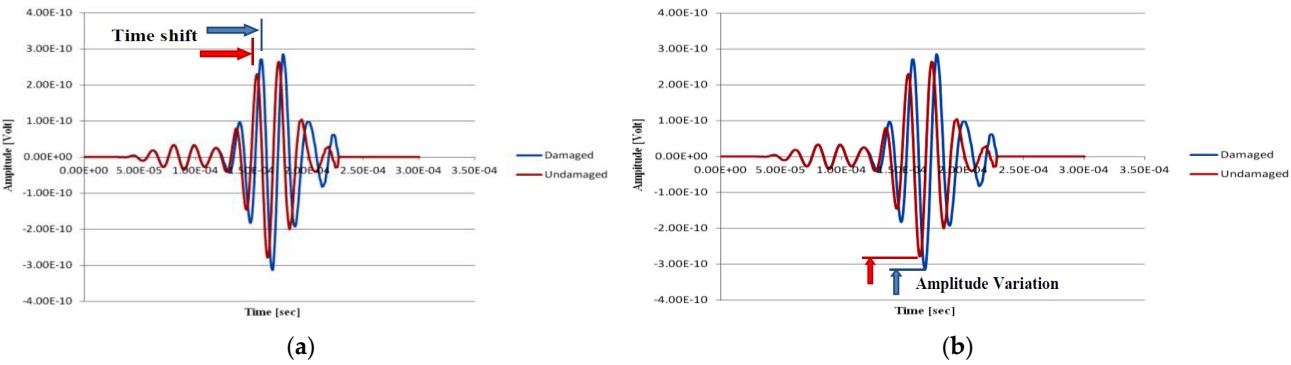

(a)                                                                   (b)

**Figure 4.** Numerical simulation of wave propagation across an impact delamination: (**a**) time shift between pristine (red) and after delamination (blue) signals; (**b**) amplitude variation between pristine (red) and after delamination (blue) signals.

Systematically, each sensor has been actuated and the signals at the other PZT locations are acquired using the classic pitch-catch method. So, the baseline signals corresponding to 156 different actuator and sensor paths for each panel bay were recorded at a known intact condition of the plate. The measurements were repeated ten times with the same methodology to characterize the collected populations with the above mentioned approach. Then, a damaged/undamaged path recognition mechanism is used to approximately locate the damage using the correlations obtained between damage index (DI), wave propagation velocities or Time of Flight (TOF) and energy level damaged/undamaged differences [9].

## 4. Signal Analysis: GW-Based Tomography and Cross-Correlation-Based Expanding Bubble Methods

Both methodologies are introduced for the comparison of SHM analysis on the tapered composite LWP under BVID impact.

### 4.1. GW-Based DI Formualtion [9]

When GW-based mathematical elaborations are carried out according to the methodology proposed in [9], the damage index (DI_GW) can be calculated as:

$$\text{DI\_GW} = \left| 1 - \frac{\sum_{f=0}^{\frac{fs}{2}} FD^2(f)}{\sum_{f=0}^{\frac{fs}{2}} FI^2(f)} \right| \tag{1}$$

where *f* are the frequencies where the spectra are evaluated, *FI* and *FD* are the magnitudes of the frequency response function for the undamaged and damaged structures, respectively and *fs* is the sample rate.

All the acquired signals, both before (pristine status) and after the impact (current status), have been treated with an STFT based script that, for each of them, calculates the ToF and the group velocity (Figure 5).

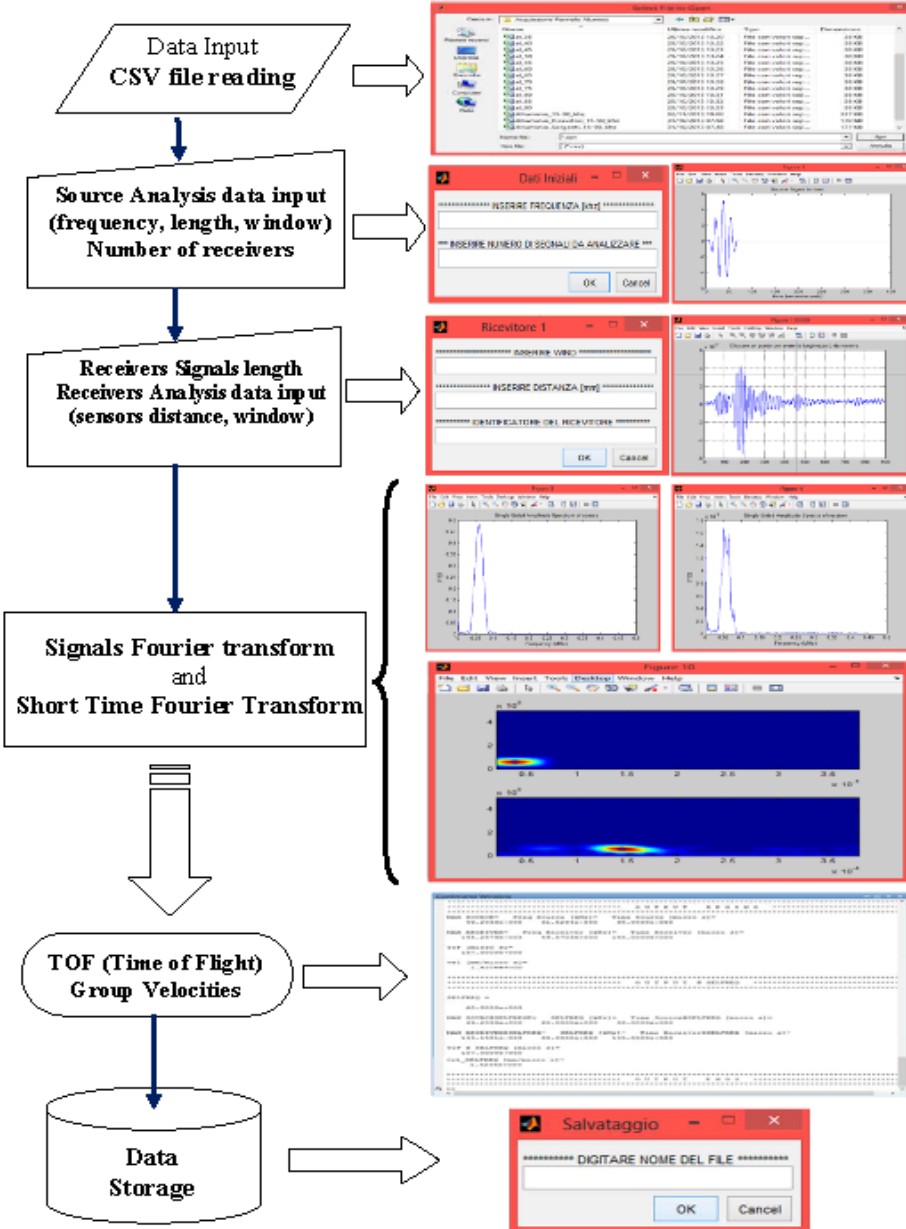

**Figure 5.** Matlab signal analysis code logical flow chart.

Code steps can be easily summarized as:

1. Signal matrix reading by CSV file. This matrix was provided directly by the acquisition system and consists of n columns of which the first is the time vector, the second the source vector and the others are the receiver's vectors;
2. Short-time Fourier Transform and the Fourier Transform calculation of the source signal and receiver's signals;
3. Determination of the source and receiver's maximum points;
4. Time of Flight (TOF) and group velocity determination.

The parameters that affect the code operations that are required in the input before analysis performing are:

- Source frequency;
- Signal length (time history length);
- Size of STFT window;
- Actuator/receiver distance.

After signal processing, the script generates a set of diagrams representative of source/receiver time histories, Fourier Transform and STFT spectrogram. In the Matlab command window, a table with ToF and group velocity calculated for each source receiver sensor couple is displayed and it is possible to save the elaboration results in an *xls* file.

Then, on the base of the saved features (ToF and group velocities), signal energy levels and damage indices in pristine and damaged statuses are compared with each other (Figure 6).

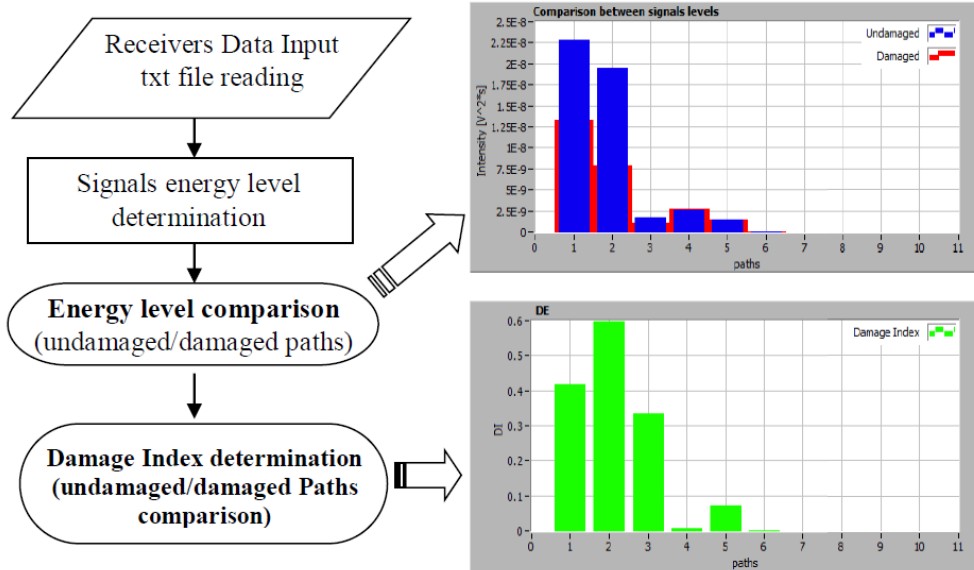

**Figure 6.** Signal levels and damage index comparison example.

Finally, to locate the damage, multiple propagation path approaches **are** adopted [9]. In this way, it is possible to identify several points forming a grid of nodes affected by the damage occurrence (see next paragraph).

### 4.2. Cross-Correlation-Based DI Formulation

For the sake of clarity, a well-known definition of cross-correlation function can be shown as:

$$R_{ij}(t) = \frac{1}{N} \sum_{l=0}^{N-1} x_i(t) x_j(t+\tau) \qquad (2)$$

where $N$ is the sample number of structural responses, $\tau$ is the time delay and when $i = j$, Equation (2) is the auto correlation function. The cross-correlation can be calculated to show how much the $i$-th signal must be anticipated along $x$-axes to make it identical to the reference $j$-th one. The formula essentially anticipates the signal along the axis, calculating the integral of the product for each possible value of the displacement.

Focusing on the specific structural response of this application, assuming that the damage is in the form of a change in stiffness, the stiffness value at position $i$ of the damaged structure can then be expressed as:

$$K_i^d = \theta_i K_i \qquad (3)$$

where $K_i$ is the stiffness of the $i$-th element in the reference state (undamaged, baseline or whatever reference status is adopted), $\theta_i$ is defined as the stiffness fraction to the reference

stiffness of the $i$-th element. $\theta = 0$ denotes that the element loses its stiffness completely, whereas $\theta = 1$ indicates that the element remains intact.

For a generic structural response under load $f$, the equation for an $N$ degree-of-freedoms (N-DOFs) viscous damped structure is given as:

$$f(t) = M\ddot{x}(t) + C\dot{x}(t) + Kx(t) \tag{4}$$

where $M$, $C$ and $K$ are the mass, damping and stiffness matrices, respectively, and $f(t)$ is the input excitation. For a static or quasi-static condition, it is demonstrated [26] that Equation (4) can be written as a function of the strain $\varepsilon(t)$:

$$B^{-T} f(t) = B^{-T} K B^{-1} \varepsilon(t) \tag{5}$$

where $B$ is proportional to the differential operator. By substituting Equation (3) in Equation (5), the response for a damaged structure can hence be simplified in Equation (6), where subscript $i$ refers to the structure element:

$$B\frac{f_i}{\theta_i K_i} = \varepsilon_i \tag{6}$$

By considering Equation (6), it is evident that a stiffener variation can be proportional to a strain variation and the cross-correlation function of Equation (2) can be written by using Equation (7) as follows:

$$R_{ij}(t) = \frac{1}{N} \sum_{l=0}^{N-1} \varepsilon_i(t)\varepsilon_j(t + \Delta\tau) \tag{7}$$

In addition, by considering a PZT electromechanical constitutive equation, linking the resulting deformation to imposed stress and electric field by a $d_{31}$ constant, the equation describing the transmitted strain at the interface between the PZT actuator and the structure may be simplified as:

$$\varepsilon_i = \frac{d_{31}}{\alpha t_p} V_i \tag{8}$$

where $\alpha$ is the generic transmission coefficient, a number in the interval [0, 1], its value depends on the geometric and physical characteristics of the materials considered. The parameter $\Lambda$ is the PZT "free" strain, or the strain that would be generated on a non-constrained piezoelectric patch, excited by an electrical tension, $V$.

From comparison of Equations (6) and (7), the cross-correlation-based damage index (DI_CC) can be calculated as [26] the function of the input voltage.

By setting the upper value of the auto-correlation envelope function of the current responses as a vector:

$$R_{max}(t) = [\max(R_{ii}(T))] \tag{9}$$

where $i = 1, 2, \ldots, n$, is the response from measurement sensor point $i$. The relative change in cross-correlation function with respect to the reference auto-correlation vector is used to define the damage index as follows:

$$\text{DI\_CC} = [(R_{ij}) - [R_{max}]] \tag{10}$$

or

$$\text{DI\_CC} = \left| \frac{\left[ \sum_{i,j=1}^{N-1} V_i(t)V_{j+1}(t + \Delta t) \right] - max\left[ \sum_{i,j=1}^{N-1} V_i(t)V_j(t + \Delta t) \right]}{N - 1} \right| \tag{11}$$

Since the cross-correlation function represents the measure of similarity of two signals, if the voltage signal at the current sensor position ($V_i$) is not affected by any variation with respect to time shift ($\Delta t$) of the coupled sensor ($V_{j+1}$), the value of the function is maximized and corresponds to the auto-correlation (max energy density) and the DI_CC is null.

All the acquired signals, both before (pristine status) and after the impact (current status), have been treated with a cross-correlation function that, for each of them, calculates the DI_CC.

The code steps can be easily summarized as:

(a) Signal matrix reading by txt file consisting of columns, of which the first is the time vector, the second the source vector and all the other the receivers' vectors (as for the GW-based approach);

(b) Cross-correlation calculation of the receivers' signal pairs;

(c) Pick track time lag estimation.

The parameters that affect the code operations and that are required in the input before analysis performing are:

- Sample rate (*time lag* $\Delta t$);
- Indexing of receivers couples.

On the base of the input signal and damage indices, the pristine and damaged status are compared with each other (Figure 7).

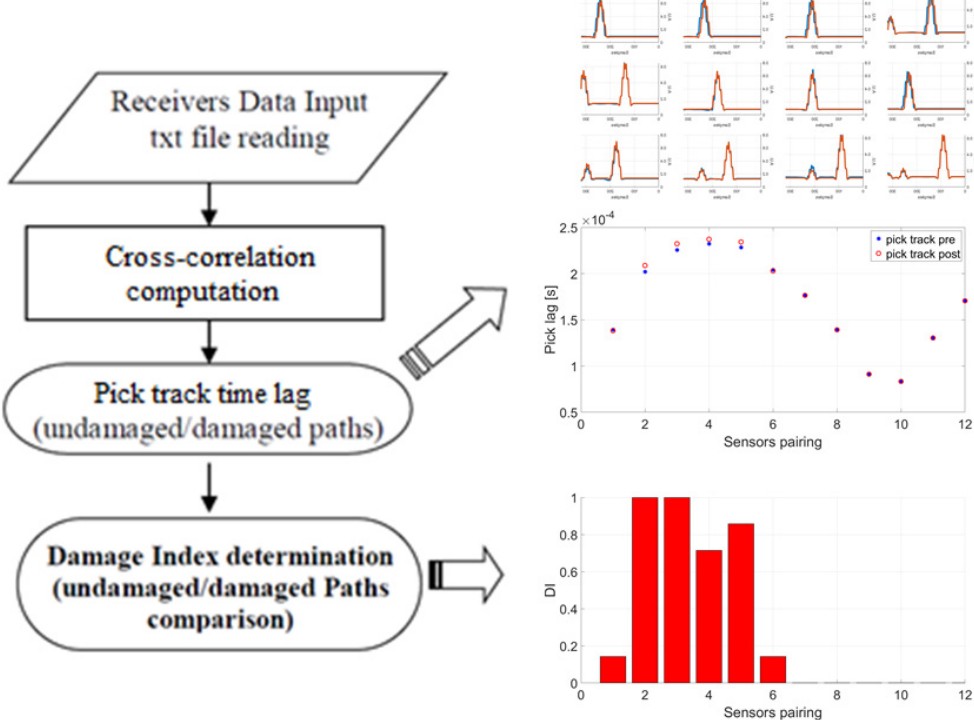

**Figure 7.** Signal levels and damage index comparison example.

The identification of the location of the damage was performed by means of a method alternative to tomography. Figure 8a depicts the geometry of the detected area used to build the schematic picture for the graphical interface shown in Figures 9 and 10. In this last one, the vertices of the polygon represent the location of the 12 outer transducers. The paths of the signal generated by them separately and detected by all the others are represented by grey lines. If a line does not cross any damage, all the intersections with other lines are represented by black dots. On the contrary, if two lines passing through the damaged area cross each other, the intersections are highlighted by red circles.

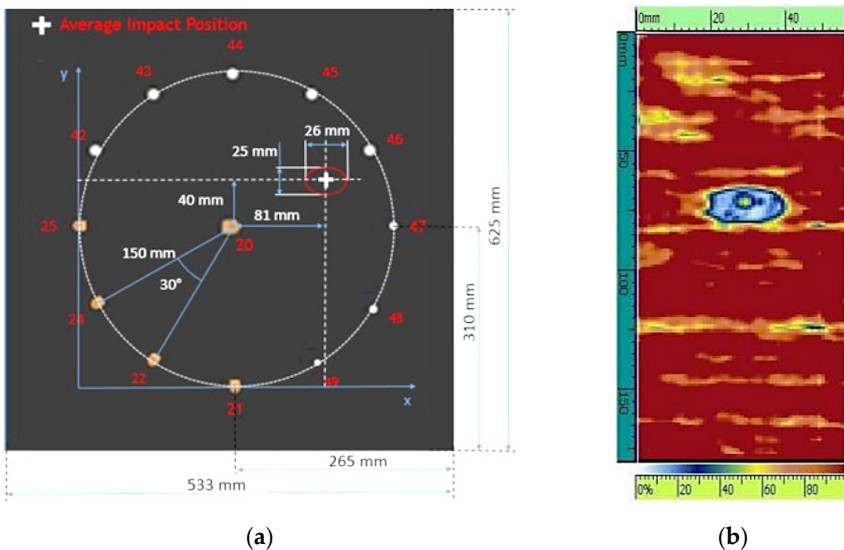

**Figure 8.** (**a**) Geometry of the panel and the damaged area; (**b**) detail of the BVID effect on the panel in the red circle.

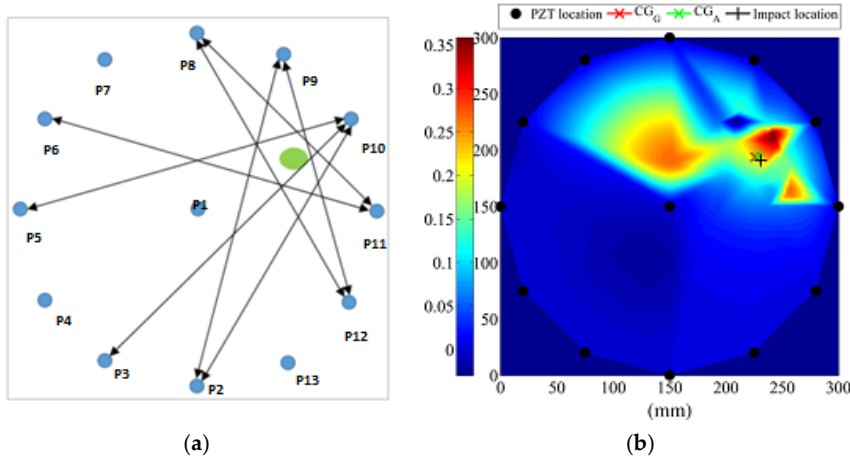

**Figure 9.** (**a**) Arrow basic GUI_GW representation of damage path and the damage position in green; (**b**) approximate shape of the damaged area by contouring [9].

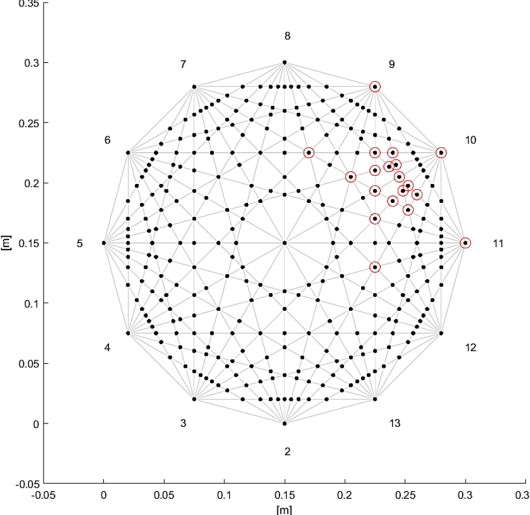

**Figure 10.** Scheme of the detected area: paths of the Lamb wave signal (gray lines); intersections of paths (bold dots); intersections of the paths on which anomaly was detected (red circles).

Finally, to locate the damage center of the mass, a modified multiple propagation path approach was adopted and presented in the next paragraph.

## 5. Results Comparison

The preliminary analysis presented in this work focused on a first test case for a damage localization SHM assessment, comparing outcomes. Briefly, in the verification study, the original SHM procedure was modified by changing code steps two to three (presented in Section 4.1), with code step b as discussed in Section 4.2.

In particular, it results in the reduction in parameters that affect the code operations and that are required in the input. Indeed, the source frequency; signal length (time history length); size of STFT window; and actuator/receiver distance can be replaced by the sample rate (*time lag* $\Delta t$) and indexing of receiver couples.

In what follows, a short description of the damaged area is provided. The PWAS were applied on the structure, following the geometry in Figure 8a. The damaged area corresponds approximately to a 26 × 25 mm ellipse, positioned in the red circle. The BVID produced an exterior surface indentation of the plate that was then inspected with C-scans. The C-scan images, reported in Figure 8b, clearly show the appearance of delamination and confirm the estimated area dimension.

The geometry of Figure 8a and the measured impact area of Figure 8b are to be used to set the input specs for the SHM graphical user interface and to provide a reference for this preliminary validation, so the results achieved by applying the DI_GW index are herein provided.

Starting from the sensor configuration (in Figure 8a), a network of propagation paths can be built considering all the possible combinations of actuator-sensor by associating a specific damage index formulation to each path (Figure 9a). Each pair of intersecting paths defines a node inside the space enclosed by the ultrasonic sensors, which contains the structural condition of its surrounding area. A contour reconstruction is then provided (Figure 9b).

Now, the results achieved by applying the DI_CC index are herein provided. A network of propagation paths can be built in a new approach:

Starting again from the sensor configuration (in Figure 8a), the paths of the guided wave signal intersections can be schematically represented as points in Figure 10. Then, the DI_CC indicates the paths on which the anomaly was revealed, and a specific algorithm is then applied to reconstruct the surface extension and barycenter estimation.

The algorithm used, called "expanding bubbles", consists of:

a. Generating growing values of the radius of circular domains centered on the light bulbs;

b. Counting the ratio of light bulbs over total dots falling in the current circular region; this value is unitary for small radii, since the center of the circles is constituted by light bulbs. On the contrary, the farther one is away from the center, the higher the probability is of finding neutral dots and, thus, the ratio tends to diminish; this trend is represented by the plot at the top of Figure 11;

c. Averaging the ratios computed at the previous step to find a unique function; the least square polynomial of this curve is then computed (see the black line in the plot at the top of Figure 11). Similarly to the single curves of the dot ratio, this line is characterized by a peak very close to the null value of the area; then after a fall, the asymptotic region starts;

d. Estimating the first derivative (slope) of the curve and its curvature (see the plots in the middle and the bottom of Figure 11). The minimum of the slope and the curvature closest to the origin identify the first part of the fall and, consequently, the boundary of the damaged area in which there is a higher concentration of light bulbs;

e. Determining of the position of the center of the damage; to this scope, the barycenter of the light bulbs is computed.

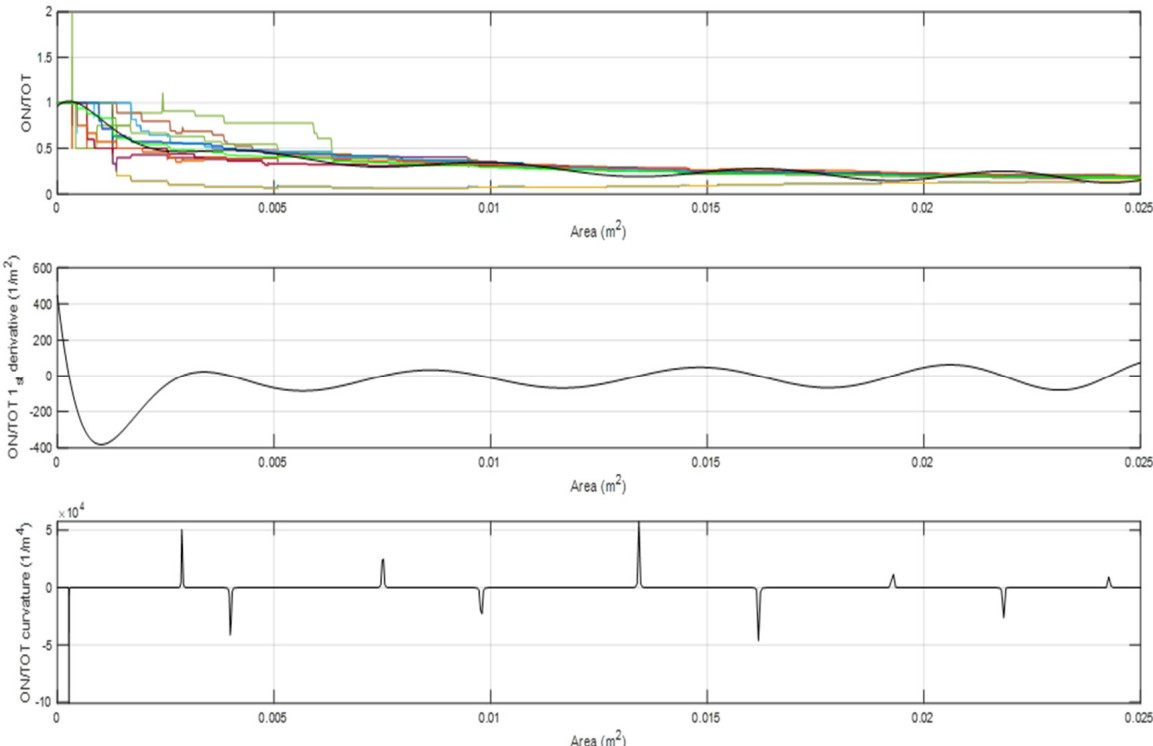

**Figure 11.** Top plot: light bulbs ratios (colored lines) and least square polynomial of the mean ratio (black bold line); middle plot: first derivative (slope) of the polynomial; bottom plot: curvature of the polynomial.

So, from the DI_CC, it is possible to associate a score to each of the PZT connected in a pitch-catch configuration. Considering the damage grids (Figure 12a), the impact locations fall exactly inside the corresponding grid.

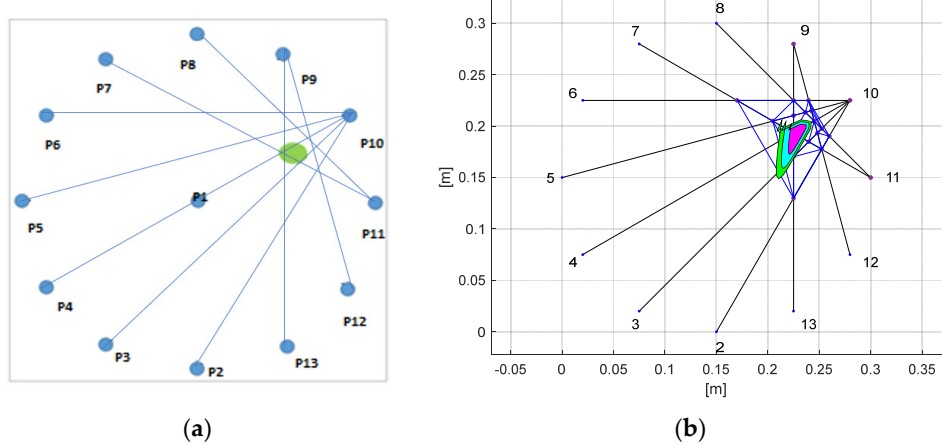

(**a**)　　　　　　　　　　　　　　　(**b**)

**Figure 12.** (**a**) Arrow basic GUI_CC representation of damaged paths and the damage in green; (**b**) approximate shape of the damaged area reconstruction.

Determining the approximate shape of the damaged area is performed by considering the boundary of the region built by a Delaunay triangulation of the light bulbs. This region is represented by the blue lines in Figure 12b. The border line is computed by the "boundary" MATLAB function, also handling concave hulls. The perimeter of the boundary is computed and scaled to achieve regions of the same shape, but with areas equal to the ones previously identified at the middle and bottom plots of Figure 11. These two regions,

the damage area at minimum curvature and slope and their mean value, are represented in Figure 12 by the pink, green and cyan patches.

By looking at the achieved results in terms of localization and barycenter estimation from Table 2, this test demonstrates the same accuracy while improving time efficiency, since the number of computational steps are reduced.

**Table 2.** Damage barycenter position.

| Damage Barycenter | $X_G$; $Y_G$ [mm] | Error $X_G$; $Y_G$ % |
|---|---|---|
| Real position | 231; 190 | ------- |
| DI_GW [9] | 227; 195 | −1.7; 2.6 |
| DI_CC | 219.6; 190.2 | −4.9; 0.1 |

## 6. Conclusions

Constantly, the V&V of SHM systems focuses only on the hardware. Indeed, current V&V methods focusing on the software are not relevant. This paper highlights the fact that these two parts need two different maturation ways and focuses on the software part by proposing a way to start assessing the outcomes.

As a conclusion of the results achieved by this work, it is important to summarize the objectives achieved. From comparison of the results obtained, applying numerical damage already developed by the University Federico II of Naples, the numerical model has been proven to work efficiently, corroborating the quality of the adopted flaw-simulating strategy.

A structural health monitoring system based on the ultrasonic guided waves (GW) tomographic technique has been presented in confirming the effectiveness of the methodology in the assessment of the structure's health condition.

However, several features and parameters that appear to be effective for damage detection can be extracted from wavefield signals and can affect SHM procedures in terms of flaw monitoring and assessment.

So, the combination of cross-correlation analysis in a data-driven approach based on an ultrasonic GW tomographic methodology could be a quick and simple tool to assess the levels of tomographic efficiency and effectiveness in the assessment of structural condition.

From this point of view, the cross-correlation method, applied as a function of a time shift of each single sensor pair, has been proven to work efficiently in seeking changes in the signal response, while the autocorrelation function has been effective for signal strain energy evaluation and damage indices definition. The provided results, indeed, are fully comparable with those obtained using the tomographic technique, confirming the integrity of the latter.

Altogether, it can be stated that satisfying steps forward have been performed in the direction of the most accurate SHM methodology. It is hopeful that further investigation will lead to more advanced operative algorithms, aiming at implementation on commercial aircrafts and allowing a relevant reduction in manufacturing, maintenance and operative costs.

The objective now is to apply this procedure in a systematic manner on each function of SHM algorithms for more mature data and damage models.

**Author Contributions:** Conceptualization, M.C. and N.D.B.; methodology, all; software, N.D.B., M.C. and S.A.; validation, M.C. and N.D.B.; formal analysis, E.M. and N.D.B.; investigation, all; data curation, E.M. and N.D.B.; writing—original draft preparation, all; writing—review and editing, all. All authors have read and agreed to the published version of the manuscript.

**Funding:** Part of the experimental results used within this work have been obtained within research funded from the European Union's Seventh Framework Programme for research, technological development and demonstration under grant agreement No. 284562 (SARISTU Project).

**Institutional Review Board Statement:** Not applicable.

**Informed Consent Statement:** Not applicable.

**Data Availability Statement:** An open database for benchmarking guided wave structural health monitoring algorithms on a composite full-scale outer wing demonstrator can be found in: *Structural Health Monitoring, 19(5), 1524–1541.*

**Conflicts of Interest:** The authors declare no conflict of interest.

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
