# Peer review of "Combination of Cross-Correlation-Based Analysis and Ultrasonic GW Tomography for Barely Visible Impact Damage Detection Preliminary Assessment"

_jcs, doi:10.3390/jcs7080321_

Round 1
Reviewer 1 Report
In the field of V&V of SHM, the main attention was paid to the hardware part of detecting defects and delaminations in laminated composites, in particular, using the ultrasound tomography method. The presented paper considers a combination of cross-correlation analysis and ultrasound tomography to determine BVID (Bulb Velocity Impact Defect ?). The authors, using the example of BVID evaluation in a full-scale model of an aircraft wing, showed the importance and effectiveness of methods for numerical processing of ultrasonic flaw detection data in composites. The localization of the defect was determined using an alternative tomography method. It is shown that the use of the cross-correlation method is very effective in searching for changes in the signal response, while the autocorrelation function is effective for estimating the signal deformation energy and determining damage indicators. The presented results are fully comparable with the results obtained using the tomographic method, which confirms its effectiveness.
This work allows us to make a significant step forward in the direction of the most accurate SHM methodology. The authors believe that further research will lead to the creation of more advanced operational algorithms designed for implementation on commercial aircraft and significantly reduce production, maintenance and operating costs.
Notes to the text. The abbreviation BVID is not deciphered anywhere in the text. (I decoded it as "Bulb Velocity Impact Defect" ???), and the SHM decoding occurs twice (lines 32, 141). The transcript of the PAS (line 185) is given after its earlier use (lines 175 and 178). The same applies to the abbreviation GW = LW (lines 92, 220, 226, 507 and 514).
The article may be published after minor amendments.
Author Response
the authors have modified the manuscript according to the reviewer suggestion.

Reviewer 2 Report
In the reviewed manuscript, the problem of impact damage localization with ultrasonic guided waves in laminate composite samples is considered. The authors examine a conventional tomography approach and propose and validate cross correlation-based expanding bubble method. For the considered experimental sample, satisfactory localization results are achieved with both of them while the latter is more simple in numerical realization and programming. The paper content seems being suitable for the Journal of Composites Science and might be of potential interest for the scholars and engineers working in the field of active SHM systems.
Some moderate revision of the manuscript text could be recommended prior its acceptance according to the following questions and suggestions related to the paper content (as they appear throughout the text):
- Since the results provided in the paper are obtained for a rather simple specimen (which, in turn, to be honest, differs rather dramatically from realistic airplane-related engineering structures), in the Introduction section, it might be suggested to pay more attention not to the importance of SHM systems and perspectives of their development but to discuss in a more comprehensive way the approaches for damage localization and visualization, especially in the sense of their practical application in the SHM software. The latter would allow, probably, to emphasize that the authors have developed relatively simple buy yet robust approach for damage localization.
- Page 2, lines 81-82. It might be suggested to provide corresponding references here.
- Table 1: Since this is a uniaxial material (probably, unidirectional carbon fiber reinforced prepreg) which is known to be transversally isotropic, how it could be that G12=G23=G31 for such a material?
- All the experiments and the results provided in Sections 3-5 are obtained for a very simple plate-like structure without any imperfections (i.e., stringers, lap-joints, etc.). In such a case, the necessity of Figures 1,2 and corresponding descriptions related to them is not clear at all and might lead only to a certain misunderstanding.
- For clarity, it might be suggested to provide wavenumber/group velocity dispersion curves of elastic guide waves propagating in the considered experimental sample.
- Page 5, lines 176-177: To what extent does such sensor placement pattern correlate with corresponding sensor locations presented in Figure 2?
- The quality of all the figures and photos should be increased. The authors could also provide any sketch of the experimental specimen where its geometry is specified.
- From the photo in Figure 3,b, it might be judged that the employed PWAS were not of the same shape? Is this a photo imperfection or are they indeed of different shape?
- Why the central frequency of 60 kHz was selected in the experiments?
- Who is the manufacturer of the employed PWAS?
- Page 10, lines 380-381: Neither black dots nor red circles are visible in Figure 8
- Page 14, line 481: What does "homonymous MATLAB function" mean?
Author Response
The authors really tjanks the reviewer for the suggestion provided in improving the qulity of the manuscript. the revision are uploaded point by point in the attached doc.

Round 2
Reviewer 1 Report
I did not point out grammatical errors in the review. The proofreader corrected them, but the caption to Figure 11 (Line 495) begins with a lowercase letter. Otherwise, the content of the article has improved significantly; it can be accepted for publication.
I did not point out grammatical errors in the review. The proofreader corrected them, but the caption to Figure 11 (Line 495) begins with a lowercase letter. Otherwise, the content of the article has improved significantly; it can be accepted for publication.
Reviewer 2 Report
The authors have carefully responded to all the comments and introduced certain improvements to the text of the manuscript. The paper could be accepted.